# Transcriptomic Analysis Reveals the Potential Mechanism of *Cardamine circaeoides* Hook.f. & Thomson in Lowering Serum Uric Acid by Reducing Inflammatory State Through CCR7 Target

**DOI:** 10.3390/ijms252312967

**Published:** 2024-12-02

**Authors:** Songrui Di, Yipeng Li, Xiangchen Fu, Chenyu Xie, Yanxin Jiang, Weican Liang, Zixu Wang, Chun Wang, Linyuan Wang, Yingli Zhu, Jianjun Zhang

**Affiliations:** 1School of Traditional Chinese Medicine, Beijing University of Chinese Medicine, Beijing 100029, China; 18331070800@163.com (S.D.); 20220931091@bucm.edu.cn (Y.L.); 20230931086@bucm.edu.cn (X.F.); xcy201116@163.com (C.X.); lwc215215@163.com (W.L.); 20230931085@bucm.edu.cn (Z.W.); chunwang_2008@163.com (C.W.); 2Institute of Basic Theory for Chinese Medicine, China Academy of Chinese Medical Sciences, Beijing 100700, China; jyx_1017@126.com; 3School of Chinese Materia Medica, Beijing University of Chinese Medicine, Beijing 100029, China; wangly@bucm.edu.cn

**Keywords:** *Cardamine circaeoides* Hook.f. & Thomson, serum uric acid, transcriptomic, molecular docking, cytokine–cytokine receptor interaction, CCR7

## Abstract

*Cardamine circaeoides* Hook.f. & Thomson (CC) is a traditional medicinal herb with multiple biological activities. In previous studies, we have identified its serum uric acid (SUA) lowering effects and speculated that *Cardamine circaeoides* water extract (CCE) may exert anti-hyperuricemia effects related to its anti-inflammatory activity. This study aims to further investigate the molecular mechanism underlying these effects at the mRNA level through transcriptomic analysis, quantitative reverse transcription polymerase chain reaction (RT-qPCR), molecular docking, and Western blotting. CCE effectively reduced SUA and improved renal function in a dose-dependent manner in hyperuricemia rats. Cytokine–cytokine receptor interaction pathway was significantly altered by CCE. An additional study identified a number of genes (IL27, Inhbe, CCR7, CXCR3, IL12RB1, CXCR5, Mstn, and GDF5) as regulators of the inflammatory response. Meanwhile, three key targets (IL27, Inhbe, and CCR7) were found to be significantly expressed at the mRNA level and have strong binding affinity with 22 components, among which Kaempferol 3-sophoroside 7-glucoside, Kaempferol-3-O-sophoroside, and Quercetin 3-sophoroside 7-glucoside have strong binding activities. Following this, Western blotting showed a significant increase in CCR7 expression. Our findings indicated that CCE regulated the cytokine–cytokine receptor interaction pathway through CCR7 to reduce the inflammatory state and exert an SUA-lowering effect.

## 1. Introduction

Hyperuricemia is a metabolic disorder characterized by abnormally high levels of SUA. Its incidence has been rising, and it is becoming increasingly common, notably among younger populations, which has attracted significant attention [1]. The clinical diagnosis of hyperuricemia is generally based on SUA levels exceeding 420 μmol/L, but treatment guidelines have proposed that clinical SUA levels ≥ 540 μmol/L or ≥480 μmol/L with comorbid conditions such as diabetes, stroke, and cardiac insufficiency should prompt pharmacological intervention [2]. Elevated SUA levels not only lead to gout but also contribute to the development of coronary heart disease, diabetes, and metabolic syndrome [3,4,5]. Therefore, controlling SUA levels within a healthy range is crucial, even for those whose SUA levels exceed 420 μmol/L but do not need pharmacological treatment. Traditional Chinese medicine (TCM) emphasizes early intervention, adopting a “prevention before disease onset” approach. Furthermore, it has also been shown to be a safe and effective way of managing SUA levels in the early stages of hyperuricemia [6].

*Cardamine circaeoides* Hook.f. & Thomson (CC) is a herb of the genus Cardamine (family Brassicaceae). Originating in Enshi, China, CC has been consumed for hundreds of years. It is a traditional medicinal herb known for its biological activities, thanks to the active compounds it contains, such as selenium-enriched proteins, polysaccharides, and peptides [7,8]. Modern pharmacology has demonstrated its anti-inflammatory and antioxidant properties in glucose and lipid metabolism [9]. In addition, no safety problems have been reported in the use of CC, and subchronic toxicity, and teratogenicity tests have found no significant toxicity [10,11]. These findings show that CC has a desirable safety profile and supports its potential for clinical use.

Previous studies have suggested that the effect of CC’s water extract (CCE) in reducing SUA was related to its anti-inflammatory mechanisms, as indicated by network pharmacology and metabolomics analyses [12]. This study aims to expand on this by exploring the mRNA-level mechanisms of CCE through transcriptomics.

Elevated SUA levels lead to an inflammatory environment in the body [13]. The cytokine–cytokine receptor interaction pathway plays an important role in both the development of hyperuricemia and the associated inflammatory response [14]. Therefore, the potential relationship between them deserves further investigation.

This study aims to investigate the potential mechanisms by which CCE reduces SUA. We employed UPLC-Q-TOF-MS for component analysis, identified key targets through transcriptomic analysis, and performed molecular docking to investigate the possibility of the interaction between them. In addition, biological validation was performed using RT-qPCR and Western blotting techniques. It is hoped that this study can clarify the mechanism of CCE in improving hyperuricemia and provide a safe and effective plant-based treatment for lowering SUA.

## 2. Results

### 2.1. Chemical Composition Identification of CCE

UPLC-Q-TOF-MS identified 22 compounds in CCE, as shown in Figure 1A,B. These include amino acids, nucleosides, dipeptides, phenolic glycosides, megastigmanes, flavonoid glycosides, phenolic acids, and fatty acids. The predominant components among them are flavonoid glycosides, including Quercetin 3-sophoroside 7-glucoside Kaempferol 3-sophoroside 7-glucoside, Quercetin-3-O-sophoroside, Kaempferol-3-O-sophoroside, Hyperoside, Naringenin-7-O-glucoside, and Astragalin (Table 1).

### 2.2. CCE Exhibits Improvement in Rats with Potassium Oxonate-Induced Hyperuricemia

To verify the effectiveness of CCE, hyperuricemia rats were treated with PG (27 mg/kg/d), H-CCE (3 g/kg/d), M-CCE (1.5 g/kg/d), or L-CCE (0.75 g/kg/d) for 30 days (Figure 2A). As shown in Figure 2B, no significant body weight changes were observed across the groups.

In Figure 2C, the SUA levels of the Model group increased from day 7 to day 30. Compared to the Model group, the PG group exhibited significantly lower levels of SUA from day 14 to 30. This confirms that the hyperuricemia model was successfully established. Compared to the Model group, the H-CCE group exhibited significantly lower levels of SUA from day 14 to 30, and the L-CCE group and the M-CCE group from day 21 to 30, with a dose-dependent effect.

### 2.3. CCE Improves Liver and Renal Function in Rats with Hyperuricemia

Given that uric acid (UA) is mainly synthesized by the liver and excreted by the kidneys, we examined the pathological manifestations of CCE in these organs. As shown in Figure 3A,B, the pathological manifestations of the kidneys and livers in these groups are presented. In the Control group, the liver tissue appeared normal, while in the Model group, significant damage and inflammatory cell infiltration were observed, as indicated by the yellow arrows. The liver tissue damage in the PG group further worsened, as evidenced by local liver cell necrosis, nuclear fragmentation, or disappearance, indicated by the black arrows. CCE treatment mitigated liver tissue damage, showing only minimal inflammatory cell infiltration in the L-CCE group (yellow arrow). Mild steatosis of the liver cells were observed in the M-CCE and H-CCE groups, as indicated by the red arrows, but without inflammatory manifestations. In kidney tissue, the Control group displayed a normal structure, while protein aceous mucus was visible in the renal tubules of the Model group, as indicated by the blue arrows and significant inflammatory cell infiltration (yellow arrows). The renal tissue damage in the PG group further worsened, with some cell necrosis and fragmented nuclei, as shown by the black arrows. CCE improved renal lesions, especially in the H-CCE group. Although some proteinaceous mucus can be seen in the glomerular capsule, as indicated by the red arrow, the renal tubular structure appeared normal, and no obvious inflammatory cell infiltration was observed.

Moreover, due to the close correlation between hyperuricemia and renal function, we evaluated the levels of serum creatinine (Scr) and blood urea nitrogen (BUN). There was no difference in BUN levels among the groups, but the levels of Scr in the Model group were obviously increased compared to the Control group (Figure 3C). We found that CCE had similar efficacy to the PG group in reducing Scr levels, with dose-dependent improvement.

### 2.4. CCE Reduces Serum Inflammation in Rats with Hyperuricemia

Hyperuricemia is a metabolic inflammatory disease caused by elevated SUA levels. In a previous study, we assessed the changes in inflammatory cytokines in the kidneys in rats with hyperuricemia. We also evaluated the systemic inflammatory changes with an ELISA test (Figure 4). The results showed an increase in the levels of interleukin-1β (IL-1β), interleukin-6 (IL-6), interleukin-8 (IL-8), and intercellular cell adhesion molecule-1 (ICAM-1), while each dose of CCE treatments was found to reduce these inflammatory cytokines, reaching levels similar to the PG group. Notably, the H-CCE treatment was able to decrease C-reactive protein (CRP) levels to a similar extent as the PG group. Furthermore, the PG group was able to reduce the elevated tumor necrosis factor-α (TNF-α) levels in the Model group. For TNF-α, although there was no significant difference in CCE treatment, a decreasing trend can be observed.

### 2.5. CCE Alters the Renal Transcriptome and Regulates the Genes Involved in Cytokine–Cytokine Receptor Interaction

To explore the underlying mechanisms of CCE in reducing hyperuricemia in rats, we performed transcriptomic analysis on renal tissue. The H-CCE group was selected as the representative intervention group for transcriptomic analysis and in-depth mechanism research. As shown in Figure 5A, a total of 176 differentially expressed genes were identified in the Model group compared to the Control group, with 72 genes being upregulated and 104 downregulated. In contrast, 179 differentially expressed genes in total were found in the H-CCE group, with 90 being upregulated and 89 downregulated. Figure 5B displays heatmaps of the differentially expressed genes that were considered targets for the occurrence of hyperuricemia and potential therapeutic targets of CCE in this study.

Next, we conducted functional and pathway enrichment analysis on the predictive targets involved in hyperuricemia and the SUA-lowering effect of CCE. In Gene Ontology (GO) functional enrichment analysis, we found significant enrichment in 402 terms and 345 terms, respectively. GO analysis revealed that differentially expressed genes associated with hyperuricemia were primarily enriched in cellular components of neuron projection, alpha–beta T-cell receptor complexes, and myosin filaments. Molecular function enrichment primarily involved DNA-binding transcription factor activity and transmembrane signaling receptor activity. Regarding biological processes, these targets were predominantly involved in the regulation of the toll-like receptor signaling pathway, lipid oxidation, and heterophilic cell–cell adhesion via plasma membrane cell adhesion (Figure 5C). The differentially expressed genes of CCE treatment were mainly enriched in cellular components, such as the extracellular space, dendrite cytoplasm, and microtubules, and mainly involved in calcium ion binding, microtubule motor activity, and signaling receptor binding as molecular functions. Regarding biological processes, these targets were primarily involved in the positive regulation of mRNA expression, positive regulation of microglial cell activation, and innate immune response.

As shown in Figure 5D, Kyoto Encyclopedia of Genes and Genomes (KEGG) pathway analysis indicated that differentially expressed genes of hyperuricemia were basically enriched in cytokine–cytokine receptor interaction, Th-cell differentiation, T-cell receptor signaling pathway, and chemokine signaling pathway. CCE treatment altered the renal transcriptome of hyperuricemia and regulated genes involved in cytokine–cytokine receptor interaction, arachidonic acid metabolism, and the TGF-beta signaling pathway. Overall, the cytokine–cytokine receptor interaction pathway may be a key pathway for CCE to reduce SUA levels. On this pathway, eight genes were enriched, namely, interleukin 27 (IL27), inhibin beta E (Inhbe), C-C chemokine receptor type 7 (CCR7), CXC-chemokine receptor 3 (CXCR3), interleukin-12 receptor subunit beta-1 (IL12RB1), CXC-chemokine receptor 5 (CXCR5), myostatin (Mstn), and growth differentiation factor 5 (GDF5) (Figure 5E,F).

To further validate CCE targets in hyperuricemia, key cytokine–cytokine receptor interaction pathway effectors were examined by biological validation. Firstly, RT-qPCR validation was performed on the above-mentioned eight genes. As shown in Figure 6, the mRNA levels of CCR7, IL27, and Inhbe were obviously changed in the Model group compared to the Control group, while the H-CCE group was regulated compared to the Model group. It also could be seen that CCE increased IL12RB1 and reduced the CXCR5 mRNA levels. Although there was no obvious variation in the expression of Mstn or GDF5 mRNA levels, it was found that CCE has a regulatory effect on the Model group. Then, we identified three key differentially expressed genes (Inhbe, CCR7, and IL27) that played important roles and molecular docked them with the components of CCE.

To further investigate the possibility of the interaction between the pivotal targets and the components of CCE, we performed molecular docking of the 22 components of CCE with IL27, Inhbe, and CCR7. It is generally believed that when the docking energy value is less than −4.25 kcal/mol, there is a significant binding activity between the two, with strong interactions indicated by values below −7.0 kcal/mol [15]. As shown in Table 2, molecular docking confirmed that the 22 components have strong interactions with all three targets. The absolute binding energy between the compounds and the target proteins was greater than −4.25 kcal/mol, indicating that they have strong affinity and may play a pivotal role in CCE’s therapeutic effect against hyperuricemia. The intersection of the five components with the most stable binding to the three key targets shows that components Kaempferol 3-sophoroside 7-glucoside, Kaempferol-3-O-sophoroside, and Quercetin 3-sophoroside 7-glucoside all have strong binding activity, as illustrated in Figure 7.

Then, we identified three key differentially expressed genes (Inhbe, CCR7, and IL27) that play important roles and conducted Western blotting validation (Figure 8). Consistent with transcriptomics prediction and mRNA expression by RT-qPCR, CCR7 protein expression in the Model group was significantly lower than that in the Control group and increased after CCE administration. In addition, IL27 and Inhbe showed a trend of change at the protein level (Figure 8).

## 3. Discussion

Epidemiological studies have confirmed a strong association between hyperuricemia and kidney diseases [5]. Since the kidney is the primary organ for UA excretion, elevated SUA levels may result in the deposition of urate salts in the kidneys, which may compromise renal function and trigger the release of inflammatory agents, leading to pathological alterations [16]. Inflammatory reactions are among the earliest signs of renal damage caused by hyperuricemia, which is one of the many ways hyperuricemia impairs renal function [17]. Currently, potassium oxonate is the most widely used drug for establishing hyperuricemia models, providing more stable SUA values compared to other methods, such as diet-induced hyperuricemia. Therefore, we chose to establish a stable hyperuricemia model by long-term gavage of potassium oxonate. In this study, the effect of CCE in reducing SUA levels was equivalent to that of the positive control drug, and CCE improved renal function indicators (Scr and BUN) and effectively improved inflammatory infiltration of renal lymphocytes. This indicates the protective effect of CCE on the kidneys, especially when SUA levels have not reached the threshold set out in the clinical guidelines for further interventions, in which case, CCE can serve as a potential option. Moreover, as UA is synthesized in the liver, and liver metabolism is also crucial for drug processing, our findings indicate that CCE does not exacerbate liver damage. Furthermore, although no significant difference was observed, we found a trend of weight loss in rats after administration of potassium oxonate. After the intervention with CCE, the weight of rats can be increased, which, we believe, may be related to the high content of nutrients such as carbohydrates, proteins, and amino acids in CCE.

We first measured serum CRP and ICAM levels since hyperuricemia and inflammatory response are closely associated. CRP is a commonly used clinical marker that can accurately reflect the systemic inflammatory response [18,19]. When hyperuricemia occurs, CRP levels significantly increase, mostly as a result of tissue damage and associated inflammatory reactions [20]. As the key biomarker of renal inflammatory activity, ICAM is a significant adhesion molecule that stimulates the release of inflammatory chemicals and mediates the adhesion of inflammatory cells [21,22]. In a preliminary study, we studied the changes in IL-1β, IL-6, IL-8, and TNF-α in the renal tissues. In this study, serum inflammatory factors were detected to evaluate systemic inflammation. The results showed that the CRP and ICAM levels, as well as all inflammatory factors, were sensibly increased in the Model group, which were reduced by the CCE treatment, particularly in the cases of ICAM and IL-6, indicating an improved inflammatory profile.

In this study, transcriptomic analyses were employed to analyze differentially expressed genes and key targets were ultimately validated at the gene and protein levels. Transcriptomics enables a comprehensive view of mRNA expression profiles under specific conditions, aligning well with the holistic approach of TCM research [23,24]. Our analysis revealed the involvement of the cytokine–cytokine receptor interaction pathway in both the development of hyperuricemia and the CCE’s SUA-lowering effects, as indicated by the KEGG enrichment results. This pathway is significantly involved in both hyperuricemia and inflammatory responses. Increased SUA in vivo induces dysregulation of cytokine activity or cytokine receptor binding that promotes chronic inflammation [14]. Furthermore, cytokines, cytokine receptors, and their interaction are generally thought to be involved in inflammatory responses [25].

Our research findings indicate that the mechanism of CCE’s SUA-lowering effect on potassium oxyzinate-induced hyperuricemia involves modulation the of the cytokine–cytokine receptor interaction pathway. We identified eight genes enriched within this pathway, including IL27, Inhbe, CCR7, CXCR3, IL12RB1, CXCR5, Mstn, and GDF5. These results show that IL27, Inhbe, and CCR7 are significantly expressed at the mRNA level, suggesting that these three targets may be the pivotal targets for CCE to reduce SUA and alleviate inflammatory response. IL12RB1, CXCR5, Mstn, and GDF5 showed distinct mRNA expression trends, though Mstn’s mRNA expression trend was opposite to the transcriptome predictions, possibly due to the differences in mRNA expression quantification between qPCR and transcriptome sequencing, resulting in their differences in estimating changes in mRNA expression level.

Molecular docking showed that the 22 components of CCE exhibited strong binding affinities with the three core targets, with flavonoid glycosides Kaempferol-3-glucoside, Kaempferol-3-O-sophoroside, and Quercetin 3-glucoside 7-glucoside showing particularly strong binding activities, indicating that they may be the key ingredients for CCE’s anti-hyperuricemia action. UPLC-Q-TOF-MS analysis confirmed a higher presence of Kaempferol-3-O-sophoroside and Quercetin 3-glucoside 7-glucoside in CCE. Based on previous research, we conducted component analysis on two batches of CCE obtained from different sources. The results revealed that they contained the same compounds, but the content of each component varied. We speculate that this may be attributed to factors such as plant age, location, and seasonality. This study identified several flavonoid components in CCE with strong binding abilities to key targets, including Kaempferol 3-sophoroside 7-glucoside, Kaempferol-3-O-sophoroside, and Quercetin 3-sophoroside 7-glucoside. Previous network pharmacology studies have similarly shown that the SUA-lowering effects of CCE were likely mediated by flavonoids, such as hyperoside and astragalin. Flavonoids are known for their ability to lower SUA levels, as evidenced by their presence in traditional Chinese medicinal herbs such as Smilacis Glabrae Rhizoma (Tufuling) and Puerariae Lobatae Radix (Gegen), which are frequently used in clinical practice. Flavonoids are the main functional components that exert their SUA-lowering effects. The flavonoids in Smilacis Glabrae Rhizoma have been shown to reduce SUA levels by promoting uric acid excretion [26], while puerarin, also a flavonoid component, can reduce SUA levels by inhibiting the activity of XOD and regulating the production of SUA [27].

Further validation revealed that there was a significant difference in protein expression of CCR7 in both the Model group and the H-CCE group, and IL27 and Inhbe protein expression did not have a statistically significant difference, but there were some clear changes to be observed. We believe that the protein expression is not only regulated at the transcriptional level but is also affected by a variety of post-transcriptional regulatory mechanisms, including splicing, editing, degradation, and stability regulation. These processes can lead to the protein expression of Inhbe and IL27, which are not significantly different from the mRNA expression level. However, there are significant differences in mRNA and protein expression of CCR7, indicating that it may be the most critical target for reducing UA and alleviating inflammatory response.

IL27, a cytokine predominantly produced by macrophages, is highly expressed in inflammatory cells [28]. When the renal tissue undergoes inflammatory infiltration, the expression of IL27 increases, indicating a correlation with renal inflammatory response [29,30]. This study found that CCE can effectively lower IL27 expression and improve renal inflammatory status. Being a member of the TGF-β superfamily, Inhbe is thought to be extensively expressed in the liver, with potential therapeutic implications for metabolic disorders such as obesity and cardiovascular [31,32]. Interestingly, research has also demonstrated that overexpressing Inhbe in high-fat diet mice reduces insulin resistance and boosts energy expenditure [33]. Inhbe levels fall in inflammatory conditions, according to research that has employed lipopolysaccharide (LPS) to produce inflammatory cell models [34]. It is interesting to note that this study revealed that Inhbe was expressed in the kidneys as well, and pro-inflammatory factors and SUA levels were negatively correlated with the Inhbe expression.

CCR7, a chemokine receptor that binds to its ligands CCL19 and CCL21, plays an important role in immune regulation [35,36]. In a clinical study of 180 cases of coronary artery vulnerable plaques, SUA levels were negatively correlated with CCR7 expression, indicating that increased SUA was associated with decreased CCR7 expression [37]. In an animal experiment, which also used potassium oxonate to induce hyperuricemia, CCR7 mRNA expression was reduced in the colon of rats after successful modeling [38]. The results of the present research indicated that the expression of CCR7 was negatively correlated with pro-inflammatory factors, likely due to the decrease in CCR7 expression after the increase in SUA, which disrupts immune balance, leading to excessive production of inflammatory factors. CCE promotes the transcription and translation of CCR7, potentially shifting the Th1/Th2 balance toward Th2 anti-inflammatory polarization and providing immune protection [39].

In conclusion, this research highlights the crucial role of the cytokine–cytokine receptor interaction pathway and its related targets in the occurrence and development of hyperuricemia. When SUA levels elevate, cytokine receptors bind to cytokines, activating intracellular signal transduction and triggering inflammatory reactions, causing damage to the kidneys. It is worth noting that the increase in CCR7 expression promotes the restoration of immune balance, thereby reducing inflammation and slowing down damage to the kidneys. The recovery of kidney function facilitates UA excretion, thereby creating a positive feedback loop that supports long-term kidney health.

## 4. Materials and Methods

### 4.1. Preparation of CCE

CC was dried and sliced (Sichuan Tongchuang Kangneng Pharmaceutical Co., Ltd., Chengdu, China, 200701). Accurately weighed slices of dried whole grass of CC were soaked in water at a 1:20 (*w*/*w*) ratio for 30 min, then boiled and extracted twice in water at a 1:20 (*w*/*w*) ratio for 30 min for the first extraction and in water at a 1:10 (*w*/*w*) ratio for 20 min for the second. Two extracts were combined and concentrated to obtain CCE, which was stored at −20 °C for this study.

### 4.2. Analysis of the CCE Chemical Composition

To determine the composition of CCE, we adopted the method described in the previous paper for UPLC-Q-TOF-MS determination [40]. Please refer to Appendix A for specific parameter settings.

### 4.3. Experimental Design

Male SD rats (180–200 g) were purchased from Vital River Laboratory Animal Technology Co., Ltd. (Beijing, China). These rats were housed in an SPF facility with free access to food and water. After one week of acclimatization, the rats were randomly divided into 6 groups (*n* = 8): the Control group (Control); Model group (Model); Positive group (PG); Low-Dose CCE group (L-CCE); Middle-Dose CCE group (M-CCE); and High-Dose CCE group (H-CCE). The Control group was intragastrically administered an equal dose of distilled water, while the other groups were given 1000 mg/kg of potassium oxonate (Shanghai yuanye Bio-Technology Co., Ltd., Shanghai, China, Y18M11C113291). After 6 h, the PG group received 27 mg/kg allopurinolal (Huahengweike (Beijing) Technology Co., Ltd., Beijing, China, ANW369), and the L-CCE, M-CCE, and H-CCE groups were given CCE at doses of 0.75, 1.5, and 3 g/kg [12], respectively. The other two groups were given the same amount of distilled water. All medications were intragastrically administered once a day for 30 consecutive days.

During this study, body weight and SUA levels were monitored on days 0, 7, 14, 21, and 30. On the last day of administration, after a 12-hour fast, blood, liver, and kidney samples were collected from all rats from each group.

### 4.4. Serum Biochemical Assay

The blood samples were centrifuged at 3000 rpm for 12 min at 4 °C to obtain serum. SUA levels were measured on days 0, 7, 14, 21, and 30. ELISA kits (Shanghai Mlbio Co., Ltd., Shanghai, China) were utilized to determine the serum contents of IL-1β (E20240607-30206B), IL-6 (E20240607-30219B), IL-8 (E20240607-30221B), ICAM-1 (E20240607-30848B), CRP (E20240607-30078B), and TNF-α (E20240607-31063B).

### 4.5. Histopathology of Renal and Liver Tissues

The renal and liver tissues of rats were washed in normal saline and fixed in 4% paraformaldehyde. The tissues underwent dehydration in a graded ethanol solution, followed by embedding in paraffin wax, and were subsequently sliced and stained with both H&E and Masson stains for routine morphological evaluation.

### 4.6. Renal Transcriptomics Sequencing

Three kidney tissue samples were taken from the Control, Model, and H-CCE groups. Total RNA was extracted using Trizol reagent (Thermofisher, Waltham, MA, USA, 15596018) following the instructions in the manual, and its quantity and purity were analyzed with Bioanalyzer 2100 and RNA 6000 Nano LabChip Kit (Agilent, Santa Clara, CA, USA, 5067-1511); high-quality RNA samples with RIN number > 7.0 were used to construct sequencing library. Then, we performed the 2 × 150 bp paired-end sequencing (PE150) on an Illumina Novaseq™ 6000, following the recommended protocol. Raw data were generated, and low-quality data were deleted to obtain clean data for subsequent analysis.

### 4.7. Identification of the Uric Acid Lowering Effect of CCE on the Hyperuricemia Gene Induced by Potassium Oxazinate

DESeq2 was employed for differential mRNA expression analysis between the two groups. Genes between the two groups with |log FC| > 1 and *p* < 0.05 were considered differentially expressed genes. Venn diagram identified common targets of differentially expressed genes in the Model vs. Control and CCE vs. Model. The results are considered potential gene targets for CCE treatment of hyperuricemia at the transcriptome level. Then, the GO function and KEGG pathway enrichment of differentially expressed genes were analyzed.

### 4.8. GO and KEGG Enrichment Analysis

GO and KEGG analyses were conducted to understand the biological roles of differentially expressed genes. GO pathways include three parts: biological processes (BP); cellular components (CC); and molecular functions (MF). KEGG enrichment analysis identified the primary pathways involved in differentially expressed genes.

### 4.9. RT-qPCR Validation

Total RNA from renal tissue was extracted by Trizol method (Servicebio, Wuhan, China), and SweScript All-in-One RT SuperMix was used to reverse transcribe 2 μL total RNA into cDNA for qPCR (Servicebio, Wuhan, China). Real-time PCR was performed using the Transcriptome instruments in CFX Connect (Bio-Rad, CFX, Hercules, CA, USA). All primers were designed and synthesized by Wuhan Servicebio Technology Co., Ltd., Wuhan, China. The relative mRNA levels were standardized to β-actin. The changes in target mRNA expression were calculated using the Relative Quantitation 2^−ΔΔ^CT method. The primer details are shown in Appendix A.

### 4.10. Molecular Docking

Key active components were obtained from the PubChem database (https://www.ncbi.nlm.nih.gov/ (accessed on 8 July 2024)) in SDF format, and key target protein structures were downloaded from the PDB database (https://www.rcsb.org/ (accessed on 8 July 2024)). The targets were optimized using pymoll-2.1.0 to remove water molecules and small molecular ligands; hydrogenation and charge treatments were performed using AutoDock Tools-1.5.6 and saved in pdbqt format. With the key targets as the receptors and the corresponding active components as the ligands, vina 2.0 was used in Pyrx for molecular docking; the binding energy was calculated, and the results file was output. The affinity (kcal/mol) value represents the binding capacity of the two, and the lower the binding capacity, the more stable it is. Finally, the strongest three binding components were visualized and analyzed in Pymol (https://pymol.org/2/ (accessed on 11 July 2024)).

### 4.11. Western Blotting Validation

Total protein was extracted from renal tissue and isolated using SDS-PAGE, then blocked with 5% skim milk after it was transferred to the PVDF membrane. Then, the membranes were subsequently blocked with blocking solution (Beyotime Biotechnology, Shanghai, China) and incubated overnight with the designated primary antibodies at 4 °C, followed by the secondary antibodies (Beyotime Biotechnology, Shanghai, China) at room temperature for 1 h. The protein blots were visualized by the ECL imaging system (Beyotime Biotechnology, Shanghai, China) and quantified by ImageJ. The primary antibodies used include rabbit anti-GAPDH (Servicebio, Wuhan, China), rabbit anti-CCR7 (Abclonal, Wuhan, China), rabbit anti-IL27 (Abcam, Cambridge, MA, USA), and rabbit anti-Inhbe (Invitrogen, Carlsbad, CA, USA).

### 4.12. Statistical Analysis

The results were expressed as means ± standard deviation (SD) for triplicate experiments or eight rats per group. Statistical analyses were performed using SPSS 21.0, with one-way ANOVA and Tukey’s post hoc test for group comparisons. *p* < 0.05 was considered statistically significant.

## 5. Conclusions

In summary, this study showed that CCE effectively lowers SUA, likely through modulation of the cytokine–cytokine receptor interaction pathway through CCR7 targets, which restores immune balance and improves the inflammatory environment. CCR7 emerged as a crucial target, potentially modifiable through silencing or overexpression to further explore its role in the inflammatory response associated with elevated SUA, and its upstream and downstream-related target pathways can be further studied. In addition, Kaempferol 3-sophoroside 7-glucoside, Kaempferol-3-O-sophoroside, and Quercetin 3-sophoroside 7-glucoside may be the key active ingredients of CCE in the treatment of hyperuricemia, providing new insights into CC’s therapeutic material basis. Collectively, these findings supply new evidence and insight that CC is a promising herb in lowering SUA.

## Figures and Tables

**Figure 1 ijms-25-12967-f001:**
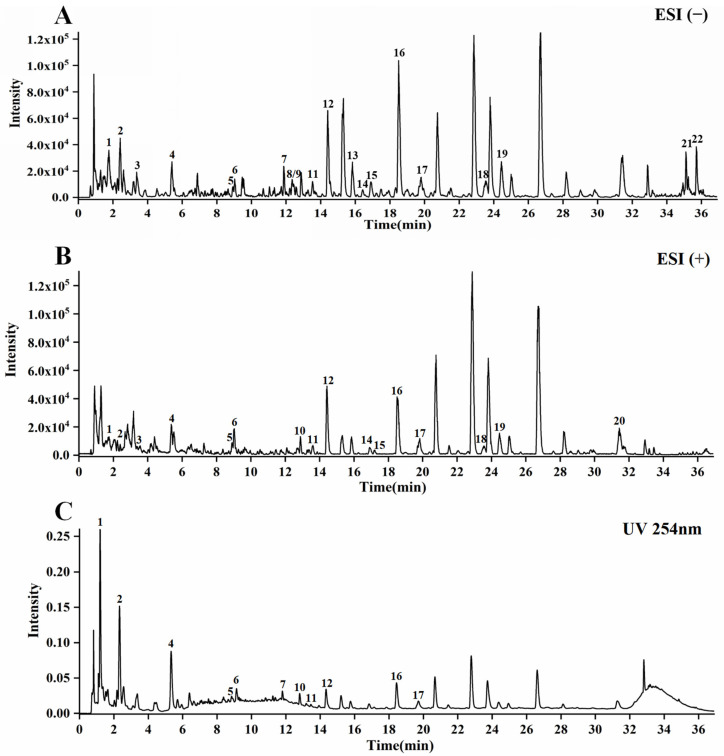
UPLC-Q-TOF-MS analysis of CCE. (**A**) ESI(−) mode. (**B**) ESI(+) mode. (**C**) UV254 nm mode. The numbers represent different components, as shown in Table 1.

**Figure 2 ijms-25-12967-f002:**
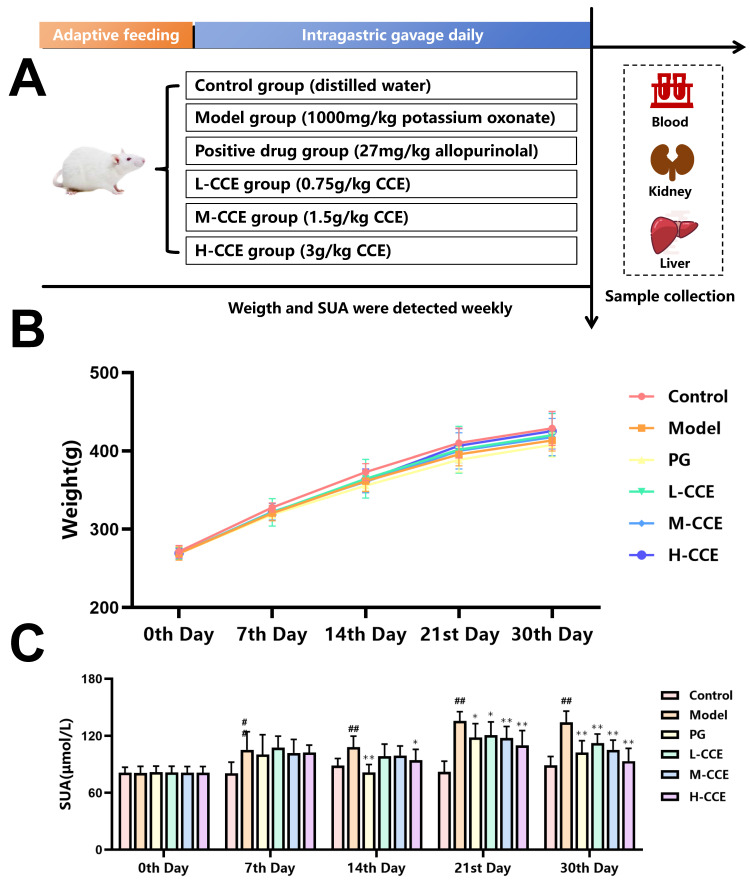
CCE alleviates the development of hyperuricemia. (**A**) Animal experiment diagram. (**B**) Body weight. (**C**) Levels of SUA. Data are shown as means ± SD (*n* = 8, ^#^
*p* < 0.05, ^##^
*p* < 0.01 vs. the Control group; * *p* < 0.05, ** *p* < 0.01 vs. the Model group).

**Figure 3 ijms-25-12967-f003:**
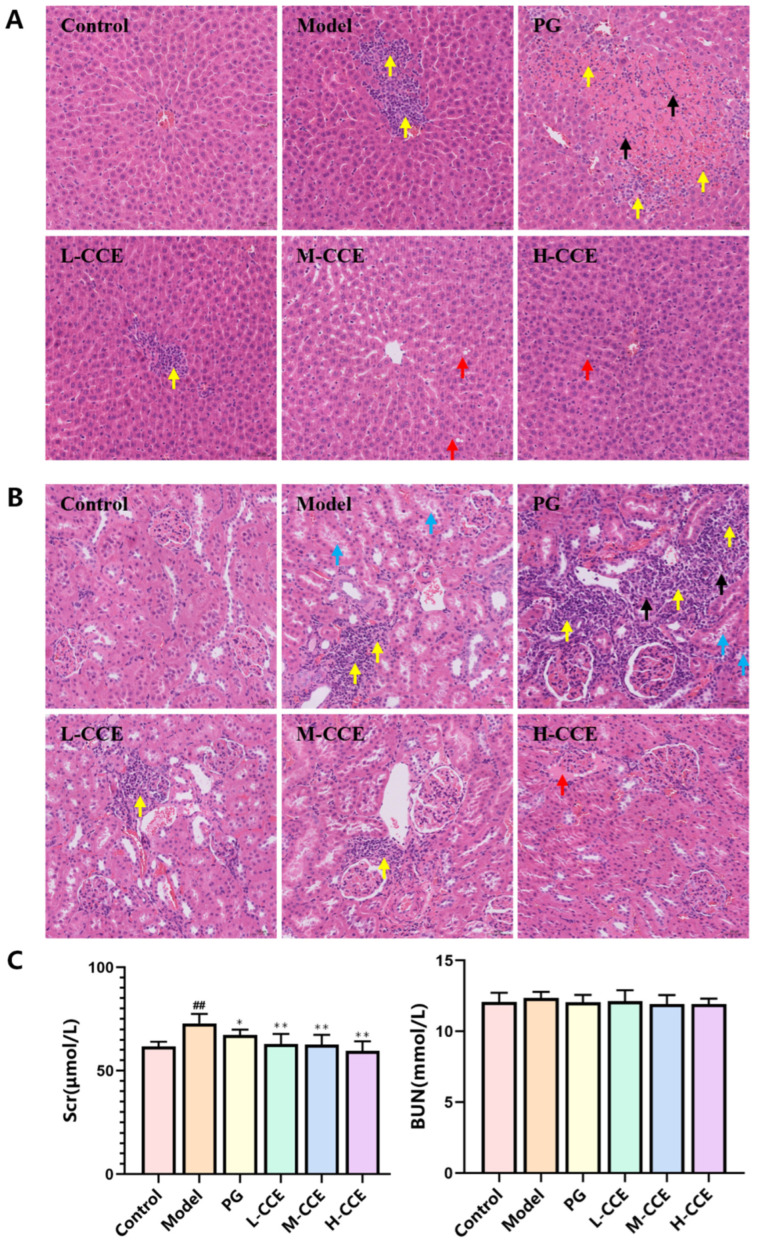
The impact of CCE on the liver and kidneys. (**A**) Liver tissue samples after hematoxylin–eosin (H&E) staining (×20). (**B**) Renal tissue samples after H&E staining (×20). (**C**) Levels of Scr and BUN. Data are shown as means ± SD (*n* = 8, ^##^
*p* < 0.01 vs. the Control group; * *p* < 0.05, ** *p* < 0.01 vs. the Model group).

**Figure 4 ijms-25-12967-f004:**
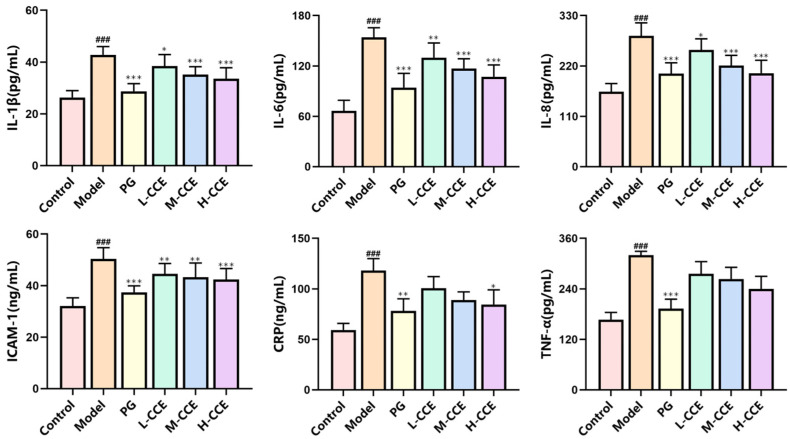
Effects of CCE on the levels of inflammatory factors in hyperuricemia rats. Data are shown as means ± SD (*n* = 8, ^###^
*p* < 0.05 vs. the Control group; * *p* < 0.05, ** *p* < 0.01, *** *p* < 0.001 vs. the Model group).

**Figure 5 ijms-25-12967-f005:**
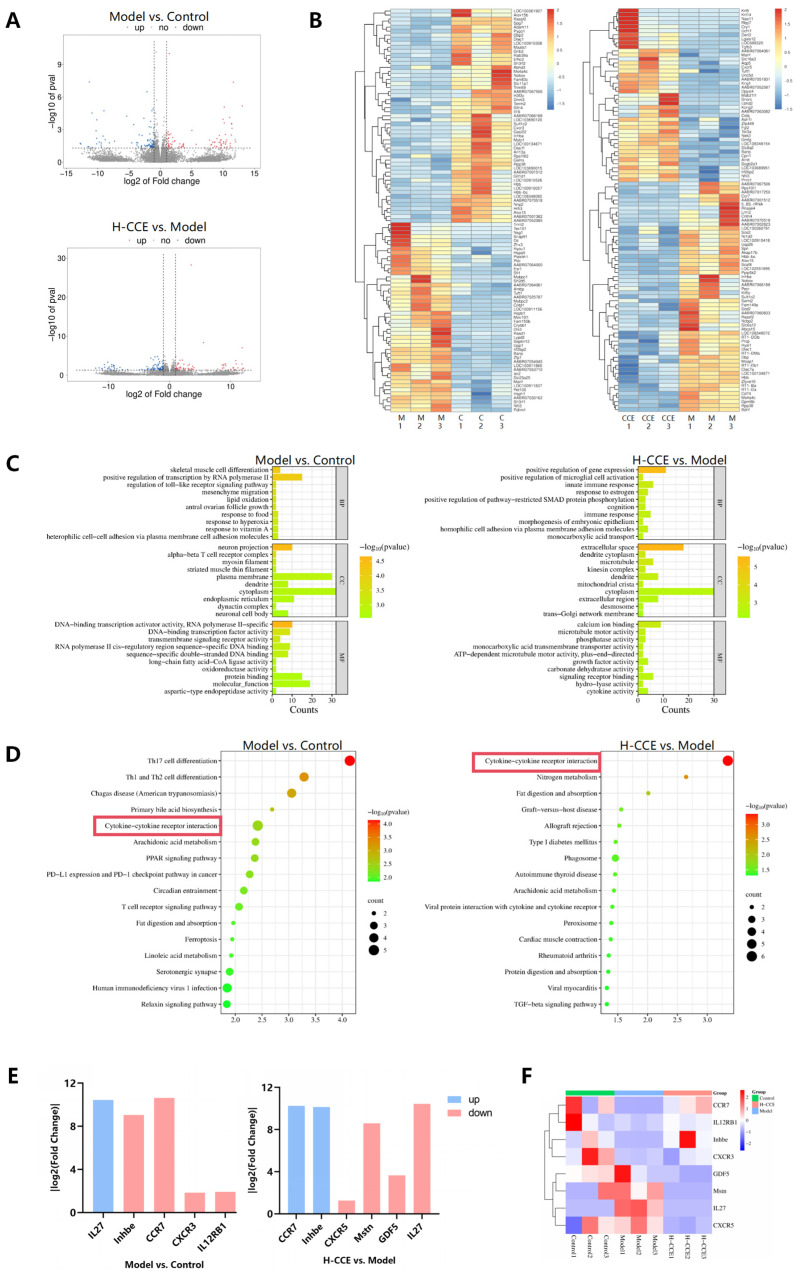
CCE had a significant effect on the renal transcriptome. (**A**) Volcano plot showing the upregulated and downregulated genes. (**B**) Cluster heatmaps display expression profiles of the significantly recovered differentially expressed genes following H-CCE treatment. (**C**) Top 10 BP, CC, and MF terms in the GO functional analysis of differentially expressed genes. (**D**) Top 20 KEGG pathways signifcantly enriched by differentially expressed genes analysis. (**E**) In the cytokine–cytokine receptor interaction pathway, target genes are significantly upregulated (blue) and downregulated (red) due to hyperuricemia or CCE treatment. (**F**) A cluster heatmap displaying the expression of differentially expressed genes enriched in the cytokine–cytokine receptor interaction pathway.

**Figure 6 ijms-25-12967-f006:**
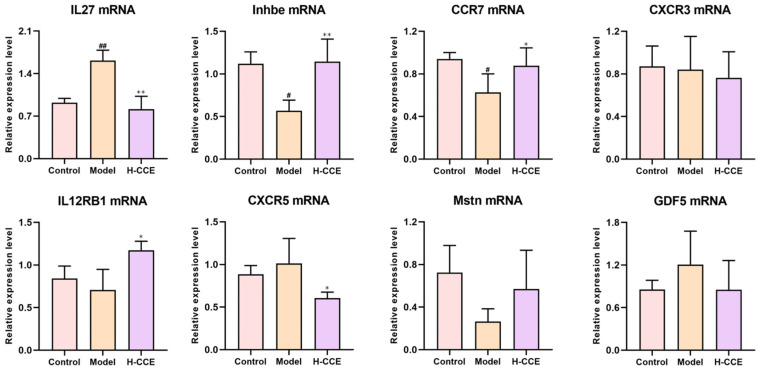
Correlative mRNA expression level of the pivotal 8 genes of CCE for the SUA-lowering effect on hyperuricemia rats. Data are shown as means ± SD (*n* = 3, ^#^
*p* < 0.05, ^##^
*p* < 0.01 vs. the Control group; * *p* < 0.05, ** *p* < 0.01 vs. the Model group).

**Figure 7 ijms-25-12967-f007:**
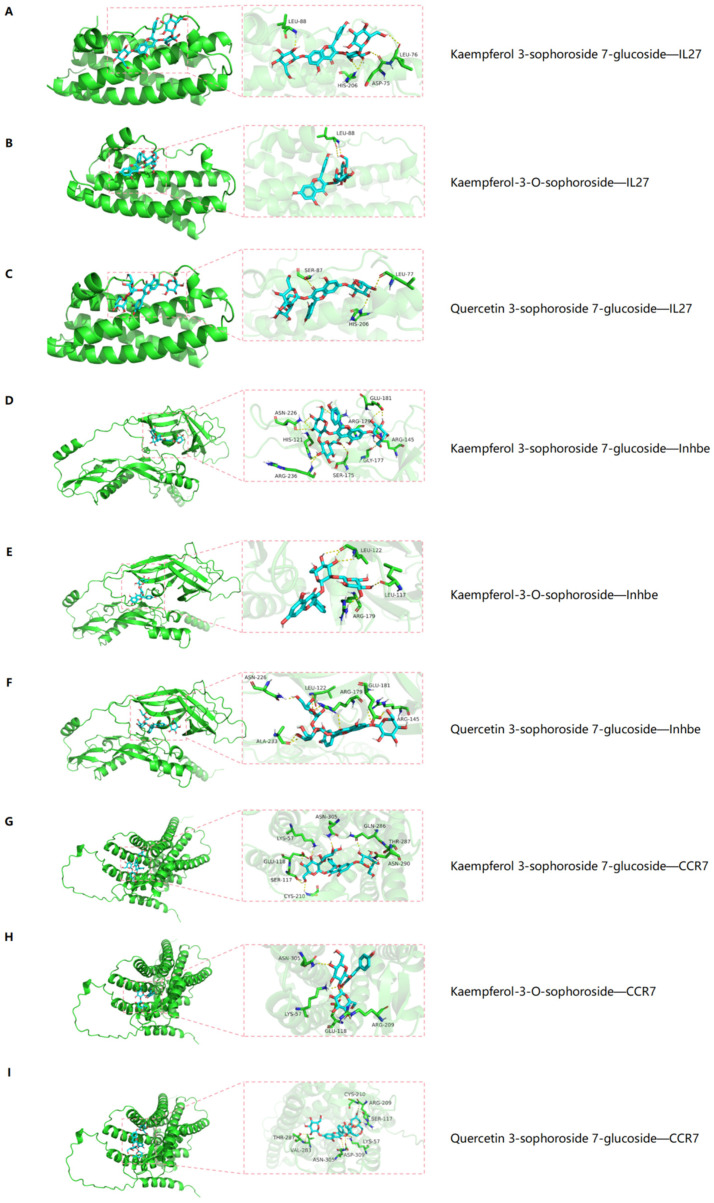
Molecular docking of the core components of CCE in reducing SUA with the key targets. Molecular docking models (Kaempferol 3-sophoroside 7-glucoside, Kaempferol-3-O-sophorosi-de, and Quercetin 3-sophoroside 7-glucoside) are shown to bind to IL27 (**A**–**C**), Inhbe (**D**–**F**), CCR7 (**G**–**I**), respectively.

**Figure 8 ijms-25-12967-f008:**
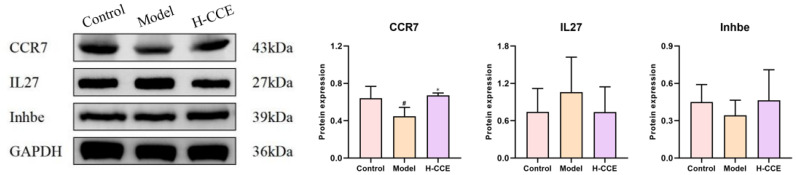
The protein expressed by CCR7, IL27, and Inhbe in renal system. Data are shown as means ± SD (*n* = 3, ^#^
*p* < 0.05 vs. the Control group; * *p* < 0.05 vs. the Model group).

**Table 1 ijms-25-12967-t001:** Ingredients identified in CCE based on UPLC-Q-TOF-MS.

No.	Time (min)		*m*/*z*	*m*/*z*	ppm	Formula	Weight	Name
1	1.93	[M − H]^−^	290.087	290.0881	−3.9	C_11_H_17_NO_8_	291.10	N-Fructosyl pyroglutamate
2	2.61	[M − H]^−^	243.0627	243.0623	1.8	C_9_H_12_N_2_O_6_	244.07	Uridine
3	3.38	[M − H]^−^	344.0405	344.0402	1	C_10_H_12_N_5_O_7_P	345.05	Cyclic guanosine monophosphate Guanosine
4	5.66	[M − H]^−^	282.0834	282.0844	−3.5	C_10_H_13_N_5_O_5_	283.09	Guanosine
5	9.08	[M − H]^−^	203.0828	203.0826	1.0	C_11_H_12_N_2_O_2_	204.09	L-tryptophan
6	9.17	[M − H]^−^	235.1081	235.1088	−3	C_12_H_16_N_2_O_3_	236.12	Phenylalanylalanine
7	11.89	[M − H]^−^	210.0782	210.0772	4.8	C_10_H_13_NO_4_	211.08	3-Methoxy-L-tyrosine
8	12.59	[M − H]^−^	385.1128	385.114	−3.2	C_17_H_22_O_10_	386.12	Sinapoylglucose
9	12.59	[M + FA − H]^−^	431.1932	431.1923	2.2	C_19_H_30_O_8_	386.19	Komaroveside A or isomer
10	12.97	[M − H]^−^	787.1938	787.1938	−0.1	C_33_H_40_O_22_	788.20	Quercetin 3-sophoroside 7-glucoside
11	13.61	[M + FA − H]^−^	449.2016	449.2028	−2.8	C_19_H_32_O_9_	404.20	Komaroveside C
12	14.55	[M − H]^−^	771.1974	771.1989	−2	C_33_H_40_O_21_	772.21	Kaempferol 3-sophoroside 7-glucoside
13	15.95	[M − H]^−^	625.1403	625.141	−1.2	C_27_H_30_O_17_	626.15	Quercetin-3-O-sophoroside
14	16.56	[M + FA − H]^−^	433.208	433.2079	0.2	C_19_H_32_O_8_	388.21	Corchoionoside A
15	16.91	[M − H]^−^	223.0615	223.0612	1.4	C_11_H_12_O_5_	224.07	Sinapic acid
16	18.60	[M − H]^−^	609.1451	609.1461	−1.7	C_27_H_30_O_16_	610.15	Kaempferol-3-O-sophoroside
17	19.81	[M − H]^−^	463.0903	463.0882	4.5	C_21_H_20_O_12_	464.10	Hyperoside
18	23.67	[M − H]^−^	433.1119	433.114	−4.9	C_21_H_22_O_10_	434.12	Naringenin-7-O-glucoside
19	24.51	[M − H]^−^	447.0917	447.0933	−3.5	C_21_H_20_O_11_	448.10	Astragalin
20	33.20	[M − H]^−^	959.2803	959.2827	−2.5	C_45_H_52_O_23_	960.29	1,2,2′-Trisinapoylgentiobioside
21	35.14	[M − H]^−^	327.2172	327.2177	−1.5	C_18_H_32_O_5_	328.23	Trihydroxyoctadecadienoic acid
22	35.74	[M − H]^−^	329.2328	329.2333	−1.7	C_18_H_34_O_5_	330.24	Trihydroxyoctadecenoic acid

**Table 2 ijms-25-12967-t002:** Binding energy of core components and key targets of CCE.

Components	Binding Energy/kcal·mol^−1^		
	IL27	Inhbe	CCR7
N-Fructosyl pyroglutamate	−6.4	−6	−6.4
Uridine	−6.3	−6.4	−6.7
Cyclic guanosine monophosphate Guanosine	−7.3	−7	−7.2
Guanosine	−6.5	−7.2	−7.2
L-tryptophan	−6.8	−6.7	−5.8
Phenylalanylalanine	−6.7	−6.3	−6.4
3-Methoxy-L-tyrosine	−5.9	−5.5	−6.2
Sinapoylglucose	−7.3	−6.8	−7.1
Komaroveside A or isomer	−7	−6.6	−7.8
Quercetin 3-sophoroside 7-glucoside	−7.6	−8.3	−9
Komaroveside C	−6.5	−6.1	−6.8
Kaempferol 3-sophoroside 7-glucoside	−8	−8	−9.1
Quercetin-3-O-sophoroside	−7.3	−8	−9.4
Corchoionoside A	−6.9	−6.5	−7.1
Sinapic acid	−6.1	−5.7	−5.8
Kaempferol-3-O-sophoroside	−7.4	−8.2	−8.8
Hyperoside	−6.7	−7.7	−7.9
Naringenin-7-O-glucoside	−8.9	−7.7	−8.7
Astragalin	−6.9	−7.4	−7.9
1,2,2′-Trisinapoylgentiobioside	−8.7	−8.5	−8.5
Trihydroxyoctadecadienoic acid	−5.1	−5.7	−5.3
Trihydroxyoctadecenoic acid	−4.9	−4.9	−5.6

## Data Availability

Data will be made available upon request.

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
