# Peer review of "Transcriptomic Analysis Reveals the Potential Mechanism of *Cardamine circaeoides* Hook.f. & Thomson in Lowering Serum Uric Acid by Reducing Inflammatory State Through CCR7 Target"

_ijms, 2024, doi:10.3390/ijms252312967_

Round 1
Reviewer 1 Report
Comments and Suggestions for Authors
The manuscript entitled "Transcriptomic analysis reveals the potential mechanism of Cardamine circaeoides Hook.f. & Thomson lowers serum uric acid by reducing inflammatory state through CCR7 target”, examines the action of the herbal extract from the Cardamine circaeoides plant in reducing uric acid in blood serum and also in improving the inflammatory state associated with hyperuricemia, utilizing molecular approaches analysis such as transcriptomic analysis, RT-qPCR, molecular docking, and Western Blotting.
The article, in my opinion, is important because it associates the plant extract with important biological actions. It could be published if some major issues are addressed:
1. Clarifying the significance of results: It is indispensable to include information on statistical significance regarding the reported data. In the article, although CCE was stated to have reduced uric acid and inflammation, statistical differences between groups should be provided-for instance, between control and model groups, and those between model and H-CCE groups. Such statistical results can either be presented in tables or graphs showing significance levels(p-value).
2. Description of cut-off values and criteria: Inflammatory marker values and measurements to assess improvement in renal function should be made clearer: for example, cutting-off values showing uric acid or inflammatory markers that define a positive response to treatment may help clarify improvements in renal function.
3. Description of the dose-response relationship: Description of the dose-response is likely by addition of information describing the decline of uric acid values with doses corresponding to L-CCE, M-CCE, and H-CCE. By this way, it becomes easier to show further responses besides peaks at different levels of that particular dose.
4. Incorporation of a comparison with allopurinol: Should allopurinol be included in this study as a positive control group? If so, it would be necessary to place additional emphasis on the act of allopurinol itself-(PG) as performed in comparison to CCE. It could be analyzed whether or not CCE covers similar indications, the comparison with allopurinol, and its advantages, limitations, and opportunities for use in place of or alongside allopurinol.
5. Highlighting the limitations of herbal remedies: Another attempt would be needed to clarify the possible advantage of CCE, such as lower toxicity and side effects. This highlights the impact that traditional medicine brings to modern treatment protocols.
6. The specific molecular mechanisms involved with the important targets: Here, the function of the key targets of CCE could be explored, CCR7 and other molecular markers (IL27, Inhbe). For instance, the increased expression of CCR7 increases the anti-inflammatory response and the alterations in CCR7 would maintain the long-term gain in kidney health.
7. Links to cytokine interactions-Netwerk: One should provide an outlook about the cytokine-cytokine receptor interaction to explain why this pathway is an important target for hyperuricemia therapeutics-more importantly in emphasis with CCR7, IL27, and Inhbe-how their impact on inflammatory conditions in the body will be highlighted.
8. Clear description of the experimental CCE extraction and analysis: It would allow explaining clearly CCE extraction methods. A short description of the extraction would function well for repeating it in further investigations.
9. Major consideration for evaluating methodology: One should define in more detail the process, that is, molecular binding, the grounds for evaluation of biological activity of the molecule, e.g. score for strong interactions binding energies-threshold value for those.
10. Uniformity in terminologies and definitions: Avoid unexplained abbreviations like differentially expressed genes (DEGs), more so on the rare possible occurrences, or some instances force their rewording towards overall comprehension of the text.
11. Correct all typos and punctuation mistakes: Rectify possible typographical errors and to improve punctuation in support of a smoother and more lucid understanding or flow of the text.
Comments on the Quality of English LanguageIt needs minor corrections of the English language.
Reviewer 2 Report
Comments and Suggestions for Authors
The study investigates the biological effects of Cardamine circaeoides extract (CCE) on hyperuricemia, a condition of elevated serum uric acid (SUA). Using various techniques including transcriptomics, RT-qPCR, molecular docking, and Western blotting.CCE significantly lowers SUA levels in a dose-dependent manner in hyperuricemic rats, comparable to the positive control drug, allopurinol. This effect is likely mediated by its anti-inflammatory properties. The study identifies that the cytokine-cytokine receptor interaction pathway plays a critical role in the inflammatory environment associated with hyperuricemia. CCE not only lowers SUA but also improves renal and liver functions, reducing inflammation in these organs. The study concludes that CCE could serve as a promising treatment for hyperuricemia by regulating the immune response and restoring balance through key pathways, particularly involving CCR7. This research highlights CCE’s potential as a plant-based remedy for hyperuricemia, providing both molecular and clinical insights into its effectiveness. Specific comments:
1. The study mentions dose-dependent effects of CCE in reducing serum uric acid (SUA) in rats. However, the study duration and specific dosages used for the different groups (L-CCE, M-CCE, H-CCE) could benefit from further elaboration. Were these doses based on a preliminary dose-response study, and if so, could that data be included or referenced?
3. Potassium oxonate was used to induce hyperuricemia in rats. While this is a common model, the study does not explain why this model was chosen over others. What are the advantages of using potassium oxonate for this study compared to other hyperuricemia models (e.g., diet-induced hyperuricemia)? Would the results differ in a different model?
4. Molecular docking analysis shows strong binding affinity of certain CCE components (e.g., Kaempferol 3-sophoroside 7-glucoside) to targets such as CCR7, IL27, and Inhbe. However, no in vivo or in vitro functional assays were performed to validate these findings. Could enzyme inhibition or receptor activation assays be conducted to experimentally validate the docking results?
5. The paper highlights eight differentially expressed genes (DEGs), but only three (IL27, Inhbe, CCR7) were chosen for RT-qPCR validation. What was the rationale for choosing these three genes for validation over others, such as CXCR3 and IL12RB1, which were also shown to be significantly affected by CCE?
6. Western blotting results for IL27 and Inhbe were reported to show a trend but not statistically significant changes, whereas CCR7 showed significant expression changes. Could the authors provide more context for why IL27 and Inhbe protein expression was not as robust as their mRNA levels? Were there issues with antibody specificity or protein degradation during the assay?
7. The study discusses histological changes in the liver and kidneys, but the data presentation could be more detailed. Could more quantitative metrics (e.g., histopathology scoring systems) be provided to objectively compare tissue damage and inflammatory cell infiltration across groups?
8. The authors focus heavily on CCR7 and its role in inflammatory responses, but the study could expand its discussion on how other immune cells (e.g., macrophages, neutrophils) are affected by CCE treatment. Were immune cell populations (e.g., macrophage or T-cell infiltration) assessed in kidney and liver tissues to further confirm the immunomodulatory effects of CCE?
10. The study provides convincing molecular evidence for the effects of CCE on hyperuricemia, but its translational potential for clinical use is not discussed in depth. How feasible is the clinical application of CCE for treating hyperuricemia in humans? Are there any toxicity or pharmacokinetic data available for CCE to support its development as a therapeutic agent?
Reviewer 3 Report
Comments and Suggestions for Authors
The paper “Transcriptomic analysis reveals the potential mechanism of Cardamine circaeoides Hook.f. & Thomson lowers serum uric acid by reducing inflammatory state through CCR7 target” investigates potential mechanisms for serum uric acid reduction in Cardamine circaeoides water extracts. The results indicated that Cardamine circaeoides water extracts has significant effect in lowering serum uric acid and its mechanism is related to regulating the cytokine-cytokine receptor interaction pathway through CCR7 targets, restoring immune balance, and improving the inflammatory environment.
The paper is interesting, the methodology is adequate and explicitly stated and the subject is very topical. The results and conclusions are remarkable and for this reason, I recommend the publication of this study after a minor revision.
Therefore, the authors are invited to clarify the following aspects:
· The novelty aspect is missing and needs to be clearly articulated. What parts do you consider original or relevant for the field? What specific gap in the field does the paper address?
· The manuscript should be checked for the possible writing errors.
· Figure 1 the compounds numbers should be written with a higher font because it is difficult to see them.
· I would suggest that in the conclusions include some final considerations on the novelties that this work provides with respect to others already existing in the bibliography.
Overall, this work is of scientific interest and is relevant within its scientific area.
Round 2
Reviewer 1 Report
Comments and Suggestions for Authors
I agree with the revised version. The authors responded to all points. I recommend publication of the manuscript without reservation.
Reviewer 2 Report
Comments and Suggestions for Authors
The study investigates the biological effects of Cardamine circaeoides extract (CCE) on hyperuricemia, a condition of elevated serum uric acid (SUA). Using various techniques including transcriptomics, RT-qPCR, molecular docking, and Western blotting.CCE significantly lowers SUA levels in a dose-dependent manner in hyperuricemic rats, comparable to the positive control drug, allopurinol. This effect is likely mediated by its anti-inflammatory properties. The study identifies that the cytokine-cytokine receptor interaction pathway plays a critical role in the inflammatory environment associated with hyperuricemia. CCE not only lowers SUA but also improves renal and liver functions, reducing inflammation in these organs. The study concludes that CCE could serve as a promising treatment for hyperuricemia by regulating the immune response and restoring balance through key pathways, particularly involving CCR7. This research highlights CCE’s potential as a plant-based remedy for hyperuricemia, providing both molecular and clinical insights into its effectiveness. The revision of the manuscript is much improved, no additional comments.
